# BDNF/TrkB Is a Crucial Regulator in the Inflammation-Mediated Odontoblastic Differentiation of Dental Pulp Stem Cells

**DOI:** 10.3390/cells12141851

**Published:** 2023-07-14

**Authors:** Ji-Hyun Kim, Muhammad Irfan, Md Akil Hossain, Anne George, Seung Chung

**Affiliations:** Department of Oral Biology, College of Dentistry, University of Illinois Chicago, Chicago, IL 60612, USA; jkim789@uic.edu (J.-H.K.); mirfan@uic.edu (M.I.); mdakil@uic.edu (M.A.H.); anneg@uic.edu (A.G.)

**Keywords:** BDNF, TrkB, dentinogenesis, DPSCs, inflammation

## Abstract

The odontoblastic differentiation of dental pulp stem cells (DPSCs) associated with caries injury happens in an inflammatory context. We recently demonstrated that there is a link between inflammation and dental tissue regeneration, identified via enhanced DPSC-mediated dentinogenesis in vitro. Brain-derived neurotrophic factor (BDNF) is a nerve growth factor-related gene family molecule which functions through tropomyosin receptor kinase B (TrkB). While the roles of BDNF in neural tissue repair and other regeneration processes are well identified, its role in dentinogenesis has not been explored. Furthermore, the role of BDNF receptor-TrkB in inflammation-induced dentinogenesis remains unknown. The role of BDNF/TrkB was examined during a 17-day odontogenic differentiation of DPSCs. Human DPSCs were subjected to odontogenic differentiation in dentinogenic media treated with inflammation inducers (LTA or TNFα), BDNF, and a TrkB agonist (LM22A-4) and/or antagonist (CTX-B). Our data show that BDNF and TrkB receptors affect the early and late stages of the odontogenic differentiation of DPSCs. Immunofluorescent data confirmed the expression of BDNF and TrkB in DPSCs. Our ELISA and qPCR data demonstrate that TrkB agonist treatment increased the expression of dentin matrix protein-1 (DMP-1) during early DPSC odontoblastic differentiation. Coherently, the expression levels of runt-related transcription factor 2 (RUNX-2) and osteocalcin (OCN) were increased. TNFα, which is responsible for a diverse range of inflammation signaling, increased the levels of expression of dentin sialophosphoprotein (DSPP) and DMP1. Furthermore, BDNF significantly potentiated its effect. The application of CTX-B reversed this effect, suggesting TrkB`s critical role in TNFα-mediated dentinogenesis. Our studies provide novel findings on the role of BDNF-TrkB in the inflammation-induced odontoblastic differentiation of DPSCs. This finding will address a novel regulatory pathway and a therapeutic approach in dentin tissue engineering using DPSCs.

## 1. Introduction

Dental caries is the most common tooth disease worldwide and is caused by the multiplex interactions of bacteria [1]. The consequence of noxious stimuli to a tooth is the formation of tertiary dentin as part of a protective repair process [2,3]. Reparative dentinogenesis is accompanied by intense inflammation that initiates stem/progenitor cell recruitment and differentiation to odontoblast-like cells. The failure of these vital processes leads to the exposure of the dental pulp and is a further cause of various infections [4]. There is a lack of evidence regarding the mechanism of action of this healing process and the significance of progenitor cell recruitment in promoting a favorable balance between inflammation and repair [5]. One possibility is that local cells release chemotactic and growth factors during the inflammation process. If this is the case, controlling inflammation to facilitate tertiary dentin formation would be a critical aspect of clinical therapy for dental caries. To date, most studies have focused on direct signaling in dentinogenesis, and only a few studies have explored the role of inflammation in this crucial process. Thus, therapeutic inflammation-induced tertiary dentin formation remains elusive. 

Dental pulp stem cells (DPSCs) are potential cells known for various regenerative medicine applications. Their original physiological function in producing odontoblasts is to create reparative dentin support applications in dentistry to regenerate tooth structures [6,7]. DPSCs are physiologically involved in the homeostasis of dentin and can be differentiated into cementoblast-like cells, collagen-forming cells with the ability to generate a cement-like material from periodontal tissue, and odontoblasts. DPSCs are also very important for maintaining the vascular and nervous homeostasis of the teeth, in addition to contributing to bone remodeling and tissue regeneration and repair [8,9]. DPSCs are derived from the neural crest, which could give rise to glia and/or neurons due to their similarities to neuronal cells and strong expression of neuronal markers like brain-derived neurotrophic factor (BDNF) and nerve growth factor (NGF), suggesting that DPSCs could actively adapt to the neuronal environment [10,11]. BDNF is a member of the neurotrophin family of growth factors and is related to the canonical NGF found in the brain and the periphery [12]. It is widely known to be involved in the regenerative process as a significant candidate molecule enhancing the formation of dentin, the outgrowth of neurites, and cell survival and death [13]. The physiological functions of BDNF include binding and signaling via the high-affinity receptor tropomyosin receptor kinase B (TrkB) and its key roles in pathophysiological effects, which are shared by immune regulators such as tumor necrosis factor (TNF) family cytokines [14].

Tumor necrosis factor alpha (TNFα) is naturally produced by activated macrophages and monocytes and has pleiotropic effects on normal and malignant cells. This significant regulatory cytokine has a role in the interaction between immune and neuronal regeneration [15]. This regulatory mediator initiates the signal transduction pathways which lead to the cellular processes of cell survival, proliferation, and differentiation. [6,16]. However, the excessive activation of TNFα signaling can lead to chronic inflammation or autoimmune diseases [17]. To avoid pathological complications, an appropriate dosage of TNFα was used in our study to enhance the odontoblastic differentiation of DPSCs and dentinogenesis. It is reported that BDNF regulates pro-inflammatory cytokines at the site of injury [18]. While the roles of BDNF in neural repair and other regeneration processes have been well identified, its role has not been explored much in the context of infection in the dentin–pulp complex. The role of BDNF and receptor-TrkB in inflammation-induced dentinogenesis remains unknown.

The aim of this research is to identify the effect of BDNF and TrkB regulation in inflammation-induced dentinogenesis in vitro. Our data show that BDNF and TrkB activation increased the TNFα-stimulated odontoblastic differentiation of DPSCs, indicating their significant role in enhancing the odontogenic differentiation of DPSCs and the regeneration of dentin. To the best of our knowledge, our proposed investigation will be the first to characterize the role of BDNF in inflammation-mediated dentinogenesis, which will leverage existing genetic approaches.

## 2. Materials and Methods

### 2.1. Chemicals and Reagents

Human dental pulp stem cells (DPSCs) were purchased from Lonza, Pharma, and Biotech (Cat. # PT-5025). The TrkB agonist (LM22A-4) was purchased from R&D System, the TrkB antagonist (Cyclotraxin-B, CTX-B) was purchased from Tocris Bioscience, and the recombinant human BDNF protein was purchased from Invitrogen. The MEMα, PBS, FBS, L-glutamine, and antibiotic–antimycotic were procured from Gibco™ Fisher Scientific (Waltham, MA, USA). Round, Poly-D-Lysine-coated (BioCoat™, 12 mm) German glass coverslips were purchased from Corning™ Fisher Scientific (Waltham, MA, USA). Various antibodies were procured: rabbit anti-BDNF (Santa Cruz, Dallas, TX, USA), rabbit anti-TrkB receptor (Proteintech, St. Louis, MO, USA), mouse anti-DMP-1 (R&D System/Sigma, St. Louis, MO, USA), and rabbit anti-DSPP (Santa Cruz, Dallas, TX, USA). Fluorescent secondary antibodies were obtained from Life Technologies (Grand Island, NY, USA). Lipoteichoic acid (LTA) from Staphylococcus aureus (Cat. # L2515) was purchased from Sigma-Aldrich (St. Louis, MO, USA).

### 2.2. Cell Culture and Differentiation

Commercially available human DPSCs isolated from the third molars of adult donors (aged 29–30 years) and collected during the extraction of wisdom teeth, which were guaranteed through 10 population doublings to express CD105, CD166, CD29, CD90, and CD73 and did not express CD34, CD45, and CD133 [19,20,21], were cultured in regular growth media (α MEM containing 10% FBS, 1% L-glutamine, and 1% antimycotic/antibiotic) at 37 °C and 5% CO_2_ for 3–4 days. The media were swapped to odontogenic media (DMEM containing 10% FBS, 1% L-glutamine, and an antimycotic/antibiotic, 50 µg/mL ascorbic acid, 10 mM β-glycerophosphate, and 10 nM dexamethasone) on day 4 until day 17. The BDNF (50 ng/mL), TrkB agonist (LM22A-4, 1 μM/mL), and TrkB antagonist (CTX-B, 200 nM/mL) were treated every three days until day 17 [22,23,24]. To induce inflammation, TNFα (20 ng/mL) and LTA were added at days 4 and 7 [25,26]. All experiments were conducted with different sets of DPSCs (between the 2nd and 4th passages) 3 times, and cell proliferation was measured by counting the total number of cells.

### 2.3. Real-Time PCR (qPCR)

The DPSCs were cultured in 6-well plate at 5 × 10^4^ cells/well, according to the differentiation protocol (Figure 1A), with the treatments BDNF, LM22A-4, or CTX-B, TNFα. The total mRNA was extracted using an RNeasy Mini Kit (Qiagen, Hilden, Germany) and analyzed using the Fisher Scientific NanoDrop 2000 device. The cDNA samples were analyzed using the Applied Biosystems SYBR green reagent system, according to the manufacture’s protocol. Primer sequences were purchased from IDT. Graph Pad Prism version 9 software was used to measure fluorescence intensity and quantification. 

Primer sequences are as follows. hGAPDH; forward: 5′-GGC ATC CAC TGT GGT CAT GAG-3′, reverse: 5′-TGC ACC ACC AAC TGC TTA GC-3′, hDSPP; forward: 5′-CTG TTG GGA AGA GCC AAG ATA AG-3′, reverse: 5′-CCA AGA TCA TTC CAT GTT GTC CT-3′, hDMP-1; forward: 5′-CAC TCA AGA TTC AGG TGG CAG-3′, reverse: 5′-TCT GAG ATG CGA GAC TTC CTA AA-3′, hRUNX-2; forward: 5′-AGA TGA TGA CAC CTC TG-3′, reverse: 5′-GGG ATG AAA TGC TTG GGA ACT-3′, hOCN; forward: 5′-CAA AGG TGC AGC CTT TGT GTC-3′, reverse: 5′-TCA CAG TCC GGA TTG AGC TCA-3′. 

### 2.4. Immunofluorescence Staining

The differentiated DPSCs were fixed and permeabilized as described previously [20]. Subsequently, the cells were incubated overnight with mouse anti-DMP-1 (1:1000), rabbit anti-DSPP (1:500), or rabbit anti-TrkB (1:1000). After the overnight incubation with primary antibodies, the secondary antibodies were treated for two hours with Alexa Fluor-594 anti-mouse IgG, Alexa Fluor-488 anti-rabbit IgG (1 μg/mL), and/or DAPI (1 μg/mL). The coverslips were mounted on the glass slides, and images were obtained using a Leica microscope. MatLab (R2022a) software was used to measure fluorescence intensity and quantification.

### 2.5. Alkaline Phosphatase Activity (ALP)

ALP activity was analyzed as an indicator of enzymatic activity consistent with mineralization. The DPSCs were cultured and fixed as previously described after 17 days of differentiation with various treatments. The cells were washed with distilled water and treated with 1 mL of AP color reagent (1% of AP color reagent A, 1% of color reagent B, and 98% of AP color development buffer) in each well. The Colorimetric AP Conjugate Substrate Kit was applied according to the manufacture’s protocol (Bio-Rad #1706432). The plates were dried completely and analyzed using a Leica DMi1 microscope to identify the enzymatic activity of extracellular matrix (ECM) mineralization in the DPSCs.

### 2.6. Alizarin Red Staining (ARS)

The 6-well plates were washed two times with distilled water and fixed with 4% PFA for one hour at RT. The cells were then washed with distilled water two times, and 1 mL of 40 mM of ARS was added per well; the ARS was provided by ScienCell (#8678). After one hour of gentle shaking, the plate was washed with distilled water (pH 4) three times and dried. The cells were inspected by using a Leica DMi1 phase microscope, and images were obtained at 20× and 40×. 

### 2.7. DMP-1 and DSPP Enzyme-Linked Immunosorbent Assay (ELISA)

Supernatants were collected from the differentiated DPSCs, and DMP-1 and DSPP ELISA kits (R&D Systems, Minneapolis, MN, USA) were used for the experiment according to the manufacturer’s protocol. A standard curve was obtained based on the standards and sample values and normalized according to the duplicated test samples at increasing concentrations.

### 2.8. In Cell Western Assay

Human DPSCs were seeded in growth media at 15 × 10^3^ cells/cm^2^ in 96-well optical-bottom plates. On day 3, the cells were treated with TNFα for 6 h. Then, the cells were immediately fixed with 100% cold methanol (15 min) and saturated with 5% BSA (1.5 h). The cells were incubated overnight at 4 °C with anti-TrkB, anti–phospho-TrkB, or anti-β-tubulin. The cells were then washed (using 0.05% Tween-20/PBS) and incubated with the respective IRDye-680RD or IRDye-800RD secondary antibody (1 h). After 5 washes, the plates were scanned at 700 and/or 800 nm (Odyssey CLx).

### 2.9. Statistical Analysis

Statistical analyses were performed using GraphPad Prism, version 9 (GraphPad Software, Boston, MA, USA), and SAS 9.4. A comparison between the two groups was carried out using an unpaired Student’s *t*-test. For multiple comparisons, the data were analyzed using a one-way analysis of variance (ANOVA), followed by the Tukey test or post-hoc Dunnett’s test to compare the different treatments and their respective controls. Data are presented as means  ±  SDs and *p* values of 0.05 or less were considered statistically significant. For the quantification of immunofluorescence staining intensity, ImageJ 1.49v software was used. Fixed areas of 1 mm × 1 mm or 2 mm × 2 mm were selected to analyze the number or fluorescence intensity of differentiated cells. Detailed statistics for each experiment are shown in the figure legend.

## 3. Results

### 3.1. DPSCs Express BDNF and TrkB

Our previous studies demonstrated that the complement C5a had a direct role in the LPS-mediated odontoblastic differentiation of DPSCs and that C5a receptors modulate the expression of BDNF, suggesting a possible linkage between the two [21,27]. So, we next investigated the role of inflammation in BDNF-mediated dentinogenesis in the context of the human odontogenic differentiation of DPSCs (Figure 1A). The DIC (Figure 1B,C) and phase contrast (Figure 1D,E) images show the proliferation and differentiation of DPSCs. Then, we first determined whether DPSCs express BDNF and its receptor TrkB. The immunofluorescent data demonstrated the expression of BDNF (Figure 1F,G) and the TrkB receptor (Figure 1H,I) on day 4. Subsequently, the treatment with a TrkB agonist confirmed an increased expression of TrkB in DPSCs (Figure 1J,K). This expression of the TrkB receptor is the first observation demonstrated in DPSCs.

### 3.2. BDNF and TrkB Agonist Significantly Enhance the Expression of Odontogenic Differentiation Markers

To determine whether DPSCs under the influence of BDNF-TrkB receptor signaling affect odontoblastic differentiation markers, we analyzed their differentiation phenotype with several assays on days 4, 7, 10, and 17. First, the pharmacological effects of BDNF and LM22A-4 on DPSCs were tested, and the expression of the odontogenic differentiation markers DSPP and DMP-1 was determined during the odontogenic differentiation of the DPSCs (Figure 2). Double immunostaining shows the expression of DSPP and DMP-1 in odontoblastic differentiated DPSCs (Figure 2A–O). The 2D line graph shows the colocalization of DSPP and DMP-1 and higher levels of expression in TrkB agonist- and BDNF-treated groups (Figure 2P–R). Figure 2S shows that the fluorescent intensities of DSPP were significantly increased in LM22A-4- (0.62 ± 0.08, *p* < 0.05) or BDNF-treated (1.03 ± 0.06, *p* < 0.001) DPSCs compared to their intensities in untreated controls (0.42 ± 0.08) at 17 days of differentiation. Similarly, DMP-1’s fluorescent intensity was observed to be higher than the control group (control 0.14 ± 0.03 vs. LM22A-4 0.34 ± 0.12, *p* < 0.05 and BDNF 0.53 ± 0.07, *p* < 0.01), indicating crucial roles of the TrkB receptor and BDNF in the odontoblastic marker’s expression.

Further, ARS staining (Figure 3A–H) was used to analyze the formation of a mineralization matrix by the DPSCs in their surrounding microenvironment during odontogenic differentiation. The results show that the mineralization activity of the DPSCs was remarkably enhanced by LM22A-4 and BDNF (control: 15.47 ± 2.71 vs. LM22A-4: 32.0 ± 6.58 *p* < 0.01 and BDNF: 29.95 ± 9.00, *p* < 0.05). Also, meaningful differences were observed between LM22A-4 or CTX-B treatments (LM22A-4: 32.0 ± 6.58 *vs.* CTX-B: 21.16 ± 6.16, *p* < 0.05) (Figure 3I).

The mRNA expression levels of the genes DSPP, DMP-1, RUNX-2, and OCN in different treatment groups of DPSCs were evaluated via qPCR to identify how the TrkB agonist and antagonist interact with DPSCs during dentinogenic differentiation. The expression of odontogenic differentiation markers was significantly upregulated in the presence of BDNF, and LM22A-4 (Figure 4A–D) compared to the control or to treatment with only CTX-B.

The result from an ELISA assay also showed that the treatment of DPSCs with BDNF or LM22A-4 increased the amount of DMP-1 in the late stage of odontogenic differentiation (Figure 5). Further, the co-application of BDNF and LM22A-4 synergistically increased the concentration of DMP-1 more than individual treatments. These data demonstrate the existence of the TrkB receptor, and the regulation of the receptor affects the early and late stages of DPSCs’ odontogenic differentiation.

### 3.3. TNFα Treatment Increases TrkB and Its Phosphorylation in DPSCs

Regenerative odontogenic differentiation associated with caries occurs in an inflammatory context. Indeed, we recently showed that the application of lipopolysaccharide (LPS), which is one of the most potent inducers of inflammation, promoted odontoblastic DPSC differentiation, as shown by the increased expression of DSPP and DMP-1 [28]. To characterize the role of BDNF-TrkB in inflammation-induced dentinogenesis, an inflammatory cytokine, TNFα was utilized. Our immunostaining data show that the fluorescent intensities of TrkB were remarkably increased in TNFα-treated DPSCs compared to untreated control DPSCs (Figure 6B–I). The in-cell Western assay shows significant increases in the expression of TrkB and its phosphorylation (Figure 6J–K). Figure 6K is a 2.5D model of Figure 6J, demonstrating increased intensities of TrkB and p-TrkB in the control and TNFα-treated groups. Figure 6L shows that the relative intensities of TrkB and p-TrkB were significantly enhanced in DPSCs treated with TNFα (TrkB and p-TrkB 357 ± 35 and 336 ± 41; *p* < 0.001) compared to those in the untreated control (TrkB and p-TrkB 100 ± 17 and 100 ± 23) DPSCs on day 4 of incubation. These data indicate that inflammation-induced dentinogenesis is associated with the activation of the TrkB receptor.

### 3.4. TNFα-Stimulated TrkB Increases the Odontoblastic Differentiation of DPSCs

Immunostaining data show that TNFα increased the expressions of odontogenic markers such as DSPP and DMP-1 in DPSCs (Figure 7). Similarly, TNFα showed a significant induction of the concentration of DSPP (control: 3.15 ± 0.08 vs. TNFα 4.57 ± 0.10 *p* < 0.05) in DPSCs via an ELISA assay. The DSPP concentration was markedly induced (3.99 ± 0.16, *p* < 0.00) by LM22A-4 but was unchanged by CTX-B compared to the concentration of DSPP in untreated control DPSCs (Figure 7Y). Co-stimulation with TNFα and LM22A-4 showed significantly increased expression levels of DSPP and DMP-1 in DPSCs (Figure 7X) compared to DPSCs treated with TNFα alone (Figure 7V), whereas treatment with both TNFα and CTX-B showed decreased expression levels of DSPP and DMP-1 (Figure 7W).

Figure 8 shows the increased mineralization activity of DPSCs via ARS (Figure 8A–H) and ALP (Figure 8I–P) treated with LTA or TNFα. A significant increase in the mineralization activity of DPSCs was observed in the ARS treated group with TNFα (31.46 ± 8.17, *p* < 0.01) compared to the untreated control (15.47 ± 2.71). It was also observed that the combined effect of TNFα and LM22A-4 was noticeably greater than the effect of TNFα treatment alone (control vs. TNFα + LM22A-4: 37.26 ± 5.12, *p* < 0.05). The results also showed that TNFα and LTA markedly induced alkaline phosphatase activity in the DPSCs which was significantly suppressed via the CTX-B treatment, indicating that TrkB regulates the mineralization stimulated by TNFα and LTA. However, TNFα treatment increased the mineralization activity by modulating TrkB, comparatively more than LTA. These results further confirm that inflammation-induced dentinogenesis is regulated via TrkB receptor.

## 4. Discussion

In this study, we defined the role of BDNF/TrkB in odontoblastic differentiation under TNFα-induced inflammation (Figure 9). Inflammation is an essential biological reaction that ensures host survival in response to infection and tissue injury. An inflammatory response is essential for the maintenance of normal tissue homeostasis [29]. Specifically, it plays a crucial role in the regeneration of injured dental tissues as it stimulates the recruitment and proliferation of pulp progenitor cells and the differentiation of odontoblasts [30,31]. The imbalance between inflammation and repair can cause irreversible damage to tissue. It has been established that inflammation is required for proficient tissue regeneration [32], yet only a few studies have investigated the role of inflammation in dentin regeneration. Our recent studies demonstrated that LTA-induced inflammation potentiates the odontoblastic differentiation of DPSCs through complement receptor C5a/p38 signaling and in vivo dentin formation was interrupted in the C5a-deficient mice [20]. Furthermore, we showed a direct relationship between C5aR receptors and BDNF modulation in pulp fibroblasts and DPSCs [27].

Neurotrophic factors are important molecules in the regulation of neuronal development and in functions such as the differentiation, maintenance, and regeneration of neurons. It is known that NGF and BDNF are mainly detected in the dental papilla/pulp and are correlated with onset of dental innervation in postnatal rodents [33]. Several studies demonstrated that BDNF’s role is significant for neurogenesis and migration [34,35]. It regulates many different cellular activities involved in the development and maintenance of normal neuronal function by binding and activating TrkB, a member of the larger family of Trk receptors [36]. BDNF is not only limited to neurons but is also discovered in many different tissue types, including the bone, cartilage, tooth germ, and heart [37]. Several studies have shown that BDNF promotes in vitro mineralization in several highly differentiated cells [38,39]. The growing evidence suggests critical roles of BDNF in tissue and bone regeneration and the migration of DPSCs. However, BDNF has not been explored much in the context of inflammation in the dentin–pulp complex. To the best of our knowledge, our data are the first to characterize the role of BDNF in dentinogenesis. Furthermore, no study has investigated the role of inflammation in BDNF-TrkB mediated dentinogenesis inDPSCs. To address this question, we set out to investigate whether the effects of BDNF enhance the odontoblastic differentiation of DPSCs subsequent to inflammation and its underlying mechanism. Our data confirm that BDNF and the activation of its receptor significantly enhance odontoblastic DPSC differentiation under inflammatory environments.

The odontoblastic differentiation of DPSCs associated with caries injury occurs in an inflammatory context [40]. Our previous article showed a link between inflammation and dental tissue regeneration, identified via dentin injury/pulp-capping in vivo [20]. In the current study, we initiated TNFα-stimulated dentinogenesis using BDNF-treated DPSCs. TNFα is responsible for a diverse range of cell-signaling events, leading to necrosis or apoptosis. We expect that TNFα could mimic the inflammatory effect in our in vitro culture condition. In our preliminary experiment, TNFα increased odontogenic markers such as DSPP and DMP-1. Furthermore, BDNF significantly potentiated its effect in the expression of DSPP and DMP-1. The application of both TNFα and CTX-B showed decreases in the expression levels of DSPP and DMP-1 compared to TNFα and BDNF treatment. To confirm the BDNF-mediated effect, we applied TNFα, BDNF, and CTX-B simultaneously and found that it better retained the expression of DSPP compared to TNFα and CTX-B treatment (Figure 7). DSPP plays a potential role in cell signaling and is known as an inducer of mineralization in the extracellular matrix [41]. It is also known that DMP-1 is expressed during (1) the early to late stages of odontogenesis via the Col1a promoter (2) or during the late stage of odontogenesis via the DSPP promoter [42]. However, the regulatory mechanism for the gene expression of DSPP and DMP-1 has not been fully revealed, especially in the inflammatory context [43]. These results identify that mild inflammation might help enhance dentinogenesis mediated by BDNF-TrkB receptor signaling. We understand that the dentin–pulp complex response to common infection is a very intricate process. It is difficult to consider all the inflammatory factors of dental caries in vitro. In this regard, bacterial and inflammatory components, i.e., LPS or cobalt (II) chloride hexahydrate, may need to be utilized to mimic the microenvironment of inflammation-induced dental caries in the future. This will possibly help us better understand the dentin–pulp complex responses in the context of carious injury. We predict that these microenvironment modifications will affect the odontogenic differentiation and its markers during the stimulation of BDNF and inflammatory factors.

While the general concepts for successful BDNF-induced regeneration have been proposed, clinical success remains difficult to achieve. For example, the clinical application of BDNF has been a major hurdle as the recombinant protein has a very short half-life (less than 10 min) that severely affects the effectiveness of this approach. In this regard, a stable and constant BDNF production platform is crucial, and stem cell engineering might fulfill this important need in the future.

## 5. Conclusions

In summary, our studies provide findings on the roles of BDNF-TrkB in inflammation-induced tertiary dentin formation and its underlying mechanism that will help to identify novel therapeutic pathways of treating dental caries.

## Figures and Tables

**Figure 1 cells-12-01851-f001:**
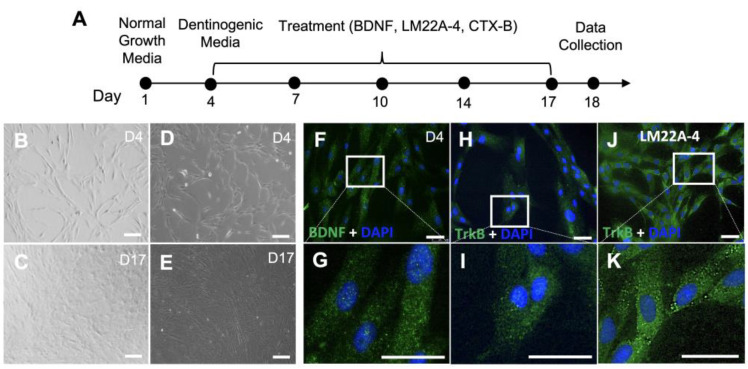
Expression of BDNF and its receptor TrkB in DPSCs. (**A**) Schematic diagram of the experiment during 17 days of the odontoblastic differentiation of DPSCs. (**B**) An image using a differential interference contrast (DIC) microscope on day 4. (**C**) A DIC image from day 17 is shown after differentiation in dentinogenic media. (**D**) Phase contrast microscopy of the DPSCs on day 4. (**E**) The morphology of the DPSCs after 17 days of odontogenic differentiation; imaged obtained via phase contrast microscopy. (**F**) Immunofluorescent image of BDNF in DPSCs on day 4. (**H**) Immunofluorescent image of TrkB receptors in DPSCs on day 4. (**J**) Immunofluorescent image of TrkB receptors in DPSCs after LM22A-4 treatment. Figure (**G**,**I**,**K**) are magnified images of Figure (**F**,**H**,**J**) (Scale bars: 50 μm).

**Figure 2 cells-12-01851-f002:**
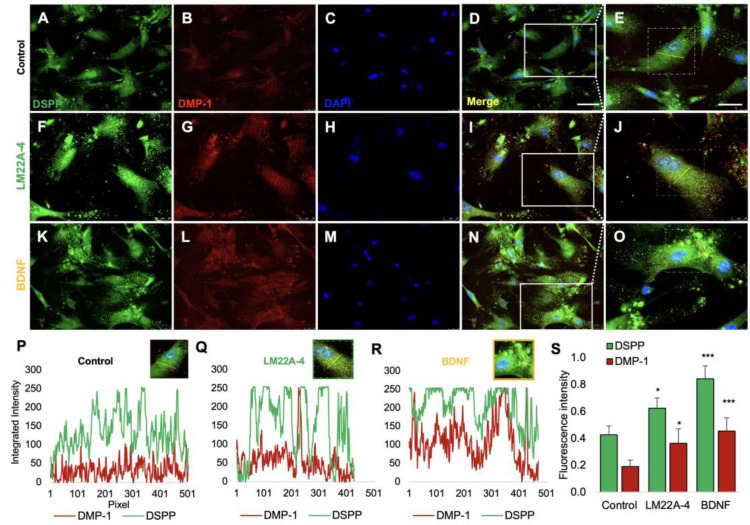
The expression of odontogenic markers DSPP and DMP-1 treated with LM22A-4 and BDNF proteins during DPSCs’ odontoblastic differentiation. (**A**) Immunofluorescent images of anti-DSPP (green) and (**B**) anti-DMP1 (red) expression during 17 days of DPSC differentiation (control); (**C**) DAPI staining after 17 days of DPSC differentiation. (**D**) Merged image of (**A**–**C**) to find co-localization of DSPP and DMP-1 in the cytoplasm of DPSCs. (**E**) Magnified image from the white box of (**D**). (**F**) Immunofluorescent images of anti-DSPP (green) and (**G**) anti-DMP1 (red) expression with the treatment of LM22A-4 on day 17. (**H**) DAPI staining. (**I**) Merged image of (**F**,**G**,**H**). (**J**) Magnified image from the white box of (**I**). (**K**) Immunofluorescent images of anti-DSPP (green) and (**L**) anti-DMP1 (red) expression with the treatment of BDNF on day 17. (**M**) DAPI staining. (**N**) Merged image of (**K**,**L**,**M**). (**O**) Magnified image from the white box of (**N**). Scale bar of (**D**,**I**,**N**): 50 μm. Scale bar (**E**,**J**,**O**): 25 μm). (**P**,**Q**,**R**) Integrated intensity of captured single cell from the dotted box (**E**,**J**,**O**). (**S**) Fluorescent intensity analysis of DSPP and DMP-1. * *p*  <  0.05, and *** *p*  <  0.001 vs. control.

**Figure 3 cells-12-01851-f003:**
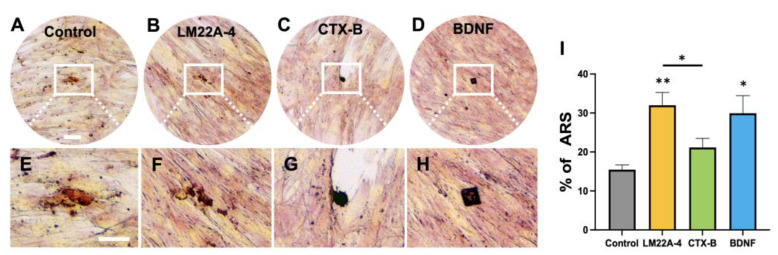
The mineralization activity of differentiated DPSCs in odontogenic media. (**A**–**D**) ARS of different treatment groups during 17 days of odontogenic DPSC differentiation. (**E**–**H**) Higher-magnification images from the white boxes of (**A**–**D**) (Scale bars: (**A**–**D**) 50 μm; (**E**–**H**) 25 μm. (**I**) The percentages of stained areas in the ARS assay. * *p*  <  0.05, and ** *p*  <  0.01 vs. control.

**Figure 4 cells-12-01851-f004:**
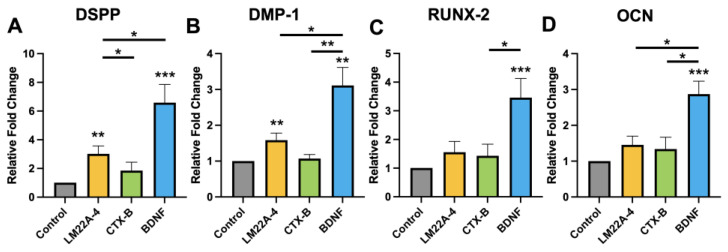
The mRNA expression of odontoblastic DPSCs treated with a TrkB agonist and antagonist. (**A**–**D**) The mRNA expression of odontogenic differentiation markers (DSPP, DMP1, RUNX-2, and OCN) during the differentiation of DPSCs with the treatment of LM22A-4, CTX-B, and BDNF. * *p*  <  0.05, ** *p*  <  0.01, and *** *p*  <  0.001 vs. control.

**Figure 5 cells-12-01851-f005:**
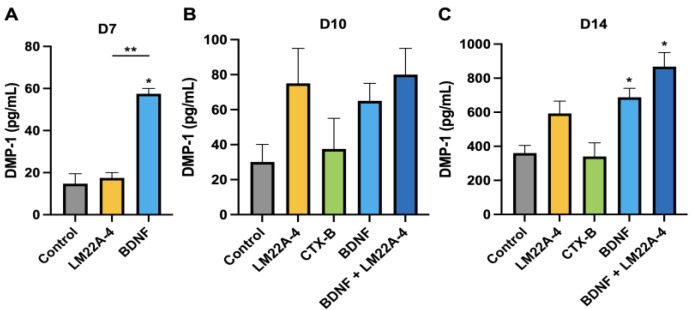
The regulation of the TrkB receptor affects the early to late stages of the odontogenic differentiation of DPSCs. A comparison of DMP-1 expression among several treatment groups at distinct time points. (**A**) Early-stage odontogenic DPSC differentiation analyzed via a DMP-1 ELISA assay; (**B**) mid-stage DMP-1 expression in DPSCs on day 10; (**C**) late-stage DMP-1 expression in DPSCs on day 14, showing significant increment in DMP-1 levels in the later stage of the odontoblastic differentiation of DPSCs. * *p*  <  0.05 and ** *p*  <  0.01 vs. control.

**Figure 6 cells-12-01851-f006:**
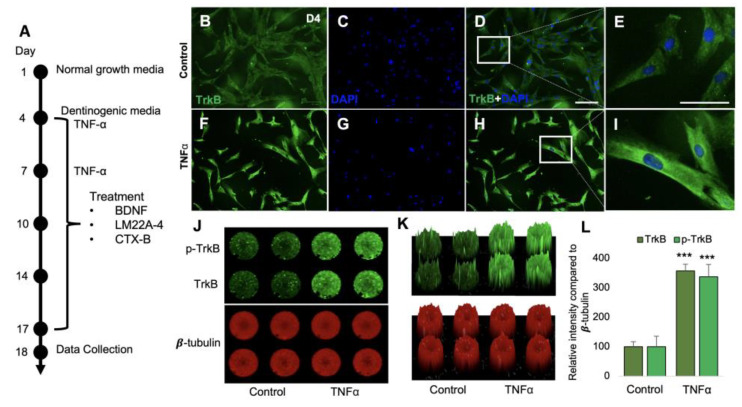
TNFα treatment increases TrkB and its phosphorylation in DPSCs. (**A**) Schematic TNFα treatment of the experiment during 17 days of odontoblastic DPSC differentiation. (**B**–**I**) Immunofluorescent image of TrkB receptors in DPSCs after TNFα treatment on day 4. (**E**,**I**) Magnified images of (**D**,**H**). (**J**) In-cell Western analysis of p-TrkB and TrkB treated by TNFα. beta tubulin was used as a housekeeping gene. (**K**) 2.5D model intensity analysis of in-cell Western assay. (**L**) The statistical analysis of in cell-Western data from (**J**,**K**). Scale bars: 50 μm. *** *p*  <  0.001 vs. control.

**Figure 7 cells-12-01851-f007:**
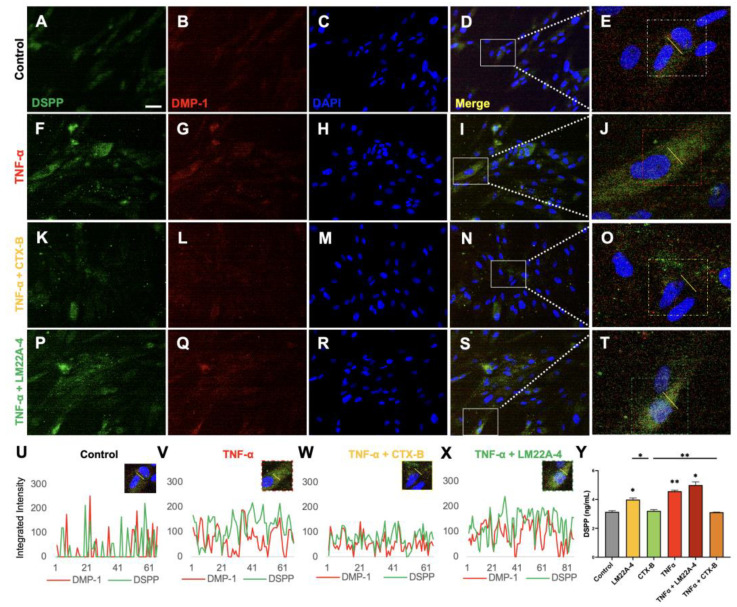
TNFα mediated by TrkB increases the odontoblastic differentiation of DPSCs. (**A**–**T**) Immunofluorescent images of DSPP and DMP-1 expression in odontogenic DPSC differentiation on day 17 (Scale bar: 50 μm). (**D**,**I**,**N**,**S**) Merged images show an overlap between DSPP (Green), DMP1 (Red), and DAPI (Blue). (**E**,**J**,**O**,**T**) Magnified images of the white box areas. (**U**–**X**) Integrated intensity line graphs of a captured single cell from the dotted box (**E**,**J**,**O**,**T**). (**Y**) The analysis of the DSPP ELISA from 17 days of odontogenic DPSC differentiation with different treatment groups. * *p*  <  0.05 and ** *p*  <  0.01 vs. control.

**Figure 8 cells-12-01851-f008:**
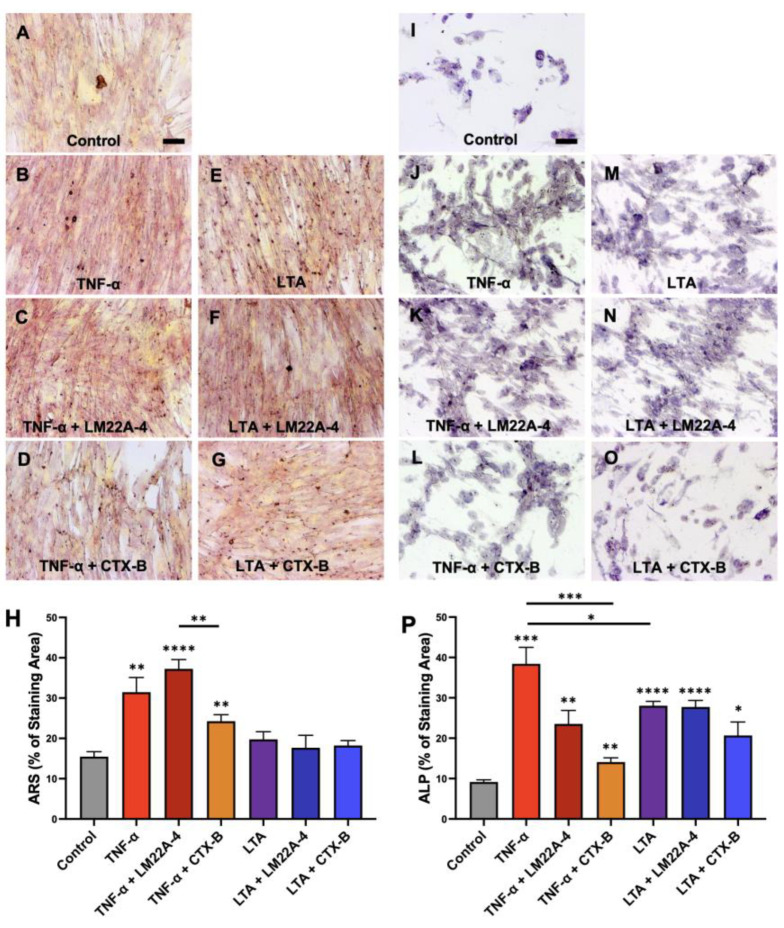
Mineralization and enzymatic activities of DPSCs treated with different inflammatory components mediated by TrkB. (**A**–**G**) ARS assay of different treatment groups during 17 days of odontogenic DPSC differentiation and the addition of inflammatory components (TNFα and LTA). (**H**) An analysis of the percentages of the stained areas in the ARS assay. (**I**–**O**) ALP assay of DPSCs with the same treatment group as ARS. (**P**) An analysis of the percentages of the stained areas in the ALP assay. Scale bars: 50 μm. * *p*  <  0.05, ** *p*  <  0.01, *** *p*  <  0.001, and **** *p*  <  0.0001 vs. control.

**Figure 9 cells-12-01851-f009:**
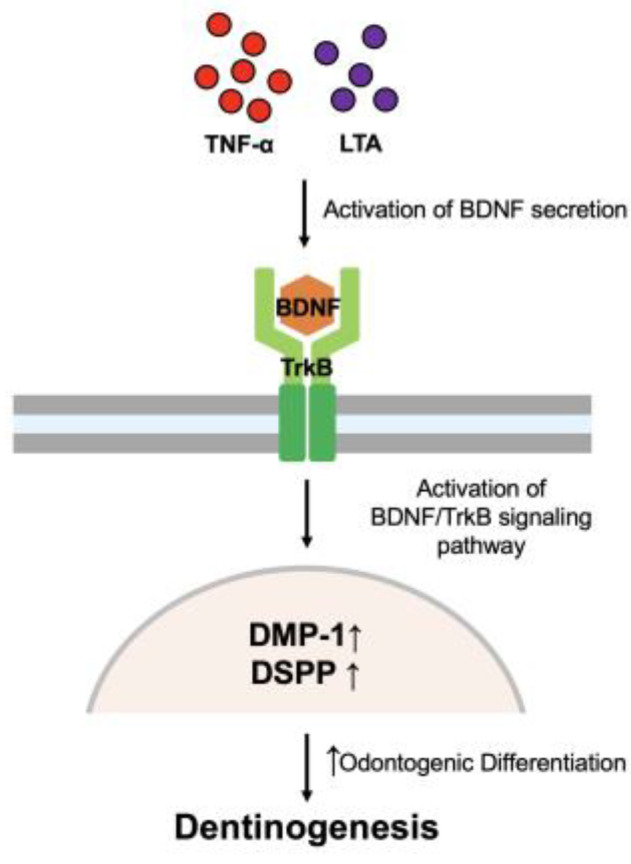
A summarized mechanism of the activation of TrkB receptor and BDNF expression during LTA- or TNFα-stimulated odontogenic DPSC differentiation.

## Data Availability

The datasets generated during and/or analyzed during the current study are available from the corresponding author upon reasonable request.

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
