# Peer review of "BDNF/TrkB Is a Crucial Regulator in the Inflammation-Mediated Odontoblastic Differentiation of Dental Pulp Stem Cells"

_cells, 2023, doi:10.3390/cells12141851_

Round 1

Reviewer 1 Report

Dear Authors, 

I read your article: "BDNF/TrkB is a crucial regulator in inflammation-mediated odontoblastic DPSCs differentiation" and I find it very interesting.

However, I suggest some things to add to improve and increase the importance of your work.

-        Check all abbreviations throughout the text, reporting the full name when first mentioned for example line 54: TNFα. 

-       In introduction, given that you are using DPSCs as an experimental model, it would be appropriate to add a brief description of these cells and their properties. I suggest some works to read and cite (DOI: 10.1073/pnas.240309797doi: 10.1080/19336896.2018.1463797; doi:10.3390/antiox10050716; doi: 10.3390/biomedicines10051056).

-        In paragraph 2 materials and methods I suggest indicating the age of the donor of the tooth from which the cells were isolated as age is significant for assessing the proliferative capacity of these cells. 

-        Check LPS and LTA among the reagents used to induce inflammation.

-        In experiments there is LTA while in materials and methods LPS.

-        Check if it is a spelling error.

-        line 90: add a reference for DPSCs characterization. 

-         Table 1: there is an error in the RUNX-2 gene sequence F and the OCN gene is incorrect.

-        Between the experiments did you make or hypothesize an intracellular measurement of Ca to confirm the differentiation?

-        Figure 1. the figure is not easy to interpret. I suggest either to divide the confocal part from the immunofluorescence part or alternatively to add the images to D17 for comparison; line 179 change G to H; line 180 change H to J; add scale bars to all images and legend.

-       line 202 change “involvement of TrkB receptor and BDNF in the expression of odontoblastic marker’s expression” in “involvement of TrkB receptor and BDNF in odontoblastic marker’s expression”.

-        Figure 2. Enhance legend sentences, from (A–O) Representative immunofluorescent images of differentiated DPSCs with BDNF and LM22A-4 protein treatment at day 17. i.e., control, LM22A-4, 208, and BDNF. It's not written clearly.

-       Figure 3. Separate in two different figures the results obtained for the mineralization activity and mRNA expression of odontogenic DPSCs differentiation. 

-        Figure 4. Why was only DMP-1 investigated in the ELISA experiment and not DSPP as well? Add the missing data or explain your choice. 

-        why didn't you use the same treatment on D7 that was used on D10 and D14?

-       In figure 6 Y The analysis of DSPP ELISA from 17 days of odontogenic DPSCs differentiation with different treatment groups why didn't you report DMP-1 too?

-        in the text starting from line 289 rewrite better the results obtained about fig.7, it is not well understood what you wanted to say.

-       line 378:  In summary, our studies will provide findings on the roles of BDNF-TrkB roles in inflammation.

Author Response

Comments from the Reviewer 1:

Dear Authors, 

I read your article: "BDNF/TrkB is a crucial regulator in inflammation-mediated odontoblastic DPSCs differentiation" and I find it very interesting. However, I suggest some things to add to improve and increase the importance of your work.

Response: Thank you so much for your keen observation and kind evaluation. We appreciate your comments and concerns, and we have tried our best to improvise the manuscript accordingly.  

(Note: Corrections/changes have been indicated with Red font in revised manuscript) 

Answers to the comments are given below:

-        Check all abbreviations throughout the text, reporting the full name when first mentioned for example line 54: TNFα. 

Response: Thank you for noticing. The full name of TNFα and more information is added in the introduction. Also, abbreviations have been double checked throughout the text.

-       In introduction, given that you are using DPSCs as an experimental model, it would be appropriate to add a brief description of these cells and their properties. I suggest some works to read and cite (DOI: 10.1073/pnas.240309797; doi: 10.1080/19336896.2018.1463797; doi:10.3390/antiox10050716; doi: 10.3390/biomedicines10051056).

Response: Thank you for your kind suggestion. We have added more details on DPSCs by citing the above-mentioned references (Line 47-58).

-        In paragraph 2 materials and methods I suggest indicating the age of the donor of the tooth from which the cells were isolated as age is significant for assessing the proliferative capacity of these cells. 

Response: Thank you for your query. Human dental pulp stem cells (DPSCs) were commercially purchased from Lonza, Pharma, and Biotech (Cat. # PT-5025); isolated from third molars of adult donors (ages 29-30 years old) collected during the extraction of wisdom teeth, which were guaranteed through 10 population doublings, to express CD105, CD166, CD29, CD90, and CD73, and do not express CD34, CD45, and CD133. We contacted the company and asked for the age of donors of the DPSCs batch delivered to us and it has been mentioned in the manuscript text (section 2.2) as well. All the experiments were conducted with different sets of DPSCs (between 2nd and 4th passages). The details have been added to methods section (Line 89-90, 103-106).

-        Check LPS and LTA among the reagents used to induce inflammation.

-        In experiments there is LTA while in materials and methods LPS.

-        Check if it is a spelling error.

Response: Thank you for the query, we used LTA and TNFα in our experiments to induce inflammation. LPS was mentioned to refer our previous studies.

-        line 90: add a reference for DPSCs characterization. 

Response: Thank you for your kind concern. References have been cited along above-mentioned corresponding details(Line 89-90, 103-106).

-         Table 1: there is an error in the RUNX-2 gene sequence F and the OCN gene is incorrect.

Response: Thank you for noticing the error. Table has been corrected and updated (Line 124).

-        Between the experiments did you make or hypothesize an intracellular measurement of Ca to confirm the differentiation?

Response: Thank you for your query. We used alizarin red staining to observe calcium deposition but did not measure the intracellular calcium. So, we illustrated based on the observation from the experimental results accordingly.

-        Figure 1. the figure is not easy to interpret. I suggest either to divide the confocal part from the immunofluorescence part or alternatively to add the images to D17 for comparison; line 179 change G to H; line 180 change H to J; add scale bars to all images and legend.

Response: Thank you for your kind concern. The figure 1 is meant for overall picture of the experiment. To provide basis, we tried to show the expression of BDNF and TrkB in DPSCs via immunofluorescence; that’s why we just shown at D4. While where there is concern of morphology, their differentiation has been shown at D4 and D17 along with experimental protocol summary. We have added the scale bars to each image for clarification.

We modified and re-arranged Figure 1 images for better understanding.     

-       line 202 change “involvement of TrkB receptor and BDNF in the expression of odontoblastic marker’s expression” in “involvement of TrkB receptor and BDNF in odontoblastic marker’s expression”.

Response: Thank you. Correction has been made accordingly (Line 213-214).

-        Figure 2. Enhance legend sentences, from (A–O) Representative immunofluorescent images of differentiated DPSCs with BDNF and LM22A-4 protein treatment at day 17. i.e., control, LM22A-4, 208, and BDNF. It's not written clearly.

Response: Thank you for your kind concern. Details have been added as suggested and figure has been modified accordingly (Line 218-227).

-       Figure 3. Separate in two different figures the results obtained for the mineralization activity and mRNA expression of odontogenic DPSCs differentiation. 

Response: Thank you for your kind suggestion. The idea here to express the results from various techniques was to show the similar pattern i.e., mineralization from ARS and mRNA expression of the related genes. Also, if we separate the images, there would be too many figures in the article. To keep the space, we merged the similar results accordingly.

Now we have re arranged and separated Figure 3 into Figure 3 and Figure 4 for better understanding according to your kind suggestion.

-        Figure 4. Why was only DMP-1 investigated in the ELISA experiment and not DSPP as well? Add the missing data or explain your choice. 

Response: Thank you for your query. While DMP-1 and DSPP both represent the mineralization activity, literature says that the DMP-1 predominantly express at 0-2 weeks (early differentiation), then DMP-1 binds to DSPP promotor, and consequently DSPP dominates between 2-4 weeks (late differentiation). Either of them could be analyzed. Considering the days, we tried to analyze (day 7, 10 and 14), we chose to quantify DMP-1 accordingly (Fig. 5 in revised version). Similarly, we checked DSPP on day 17 in Figure 7Y for the same reason, accordingly.

-        why didn't you use the same treatment on D7 that was used on D10 and D14?

Response: Thank you for your question. We initially checked whether BDNF has effect on DMP-1 as a preliminary experiment. Then, we decided to evaluate at day 10 and 14 with various treatments. Day 7 data was not supposed to be mentioned but we just decided to include it anyway.

-       In figure 6 Y The analysis of DSPP ELISA from 17 days of odontogenic DPSCs differentiation with different treatment groups why didn't you report DMP-1 too?

Response: Thank you for your kind concern. We decided to choose the DSPP in this experiment for the same above-mentioned reason.

-        in the text starting from line 289 rewrite better the results obtained about fig.7, it is not well understood what you wanted to say.

Response: Thank you for your kind concern. We have revised the results for better clarification (Line 307-308, 312-317 in revised version).

-       line 378:  In summary, our studies will provide findings on the roles of BDNF-TrkB roles in inflammation.

Response: Thank you for pointing out. We have revised the statement.

We are really thankful and appreciate your efforts to evaluate our study; it really helped us improving the quality of our manuscript and we do hope that concerned queries have been answered accordingly.  

Thanks for anticipation. 

Reviewer 2 Report

In the manuscript entitled “BDNF/TrkB is a crucial regulator in inflammation-mediated odontoblastic DPSCs differentiation”. The authors assessed the role of BDNF receptor-TrkB in the inflammation-induced dentinogenesis of DPSCs.

Comments

1.   Did the authors perform different concentrations of LM22A-4 and CTX-B to use the final optimal concentration?

2.   In the introduction, the authors mentioned that to avoid pathological complications, an appropriate dosage of TNFα was used. However, the authors did not prove that the dose of used TNF did not induce chronic inflammation.

3.   To synchronize all the cells to the same cell cycle phase, cells need to be starved before treatment. In the current study, cells were treated in a complete regular media and did not starve??

4.   In the results sections, please don’t repeat the methods.

5.   In Fig.2, S. Why authors used resting rather than control.

English language is fine

Author Response

Comments from the Reviewer 2:

In the manuscript entitled “BDNF/TrkB is a crucial regulator in inflammation-mediated odontoblastic DPSCs differentiation”. The authors assessed the role of BDNF receptor-TrkB in the inflammation-induced dentinogenesis of DPSCs.

Response to Reviewer #2,

Dear Reviewer,

Thank you for your keen observations and kind evaluations. We appreciate your comments and concerns, and we have tried our best to address them and revised the manuscript accordingly. We do hope that manuscript is acceptable in current revised form.

Response to your comments/concerns is as follows:

(Note: Changes/corrections have been highlighted with Red font in revised version of manuscript)

Comments

  1. Did the authors perform different concentrations of LM22A-4 and CTX-B to use the final optimal concentration?

Response: Thank you for your query. We used optimal concentration mentioned in literature and it worked well (Line 111-112, Ref. 23-24) 

  1. In the introduction, the authors mentioned that to avoid pathological complications, an appropriate dosage of TNFα was used. However, the authors did not prove that the dose of used TNF did not induce chronic inflammation.

Response: Thank you for pointing out and we totally agree with your kind concern. We have been regularly using TNFα, LTA and LPS in DPSCs, MSCs and fibroblasts in our lab and we have published these data in our recent previous studies. Also, literature confirms the appropriateness of the dosage we used (Line 112-114, Ref. 25-26). Please refer to the following references.

Chmilewsky, Fanny, et al. "C5L2 regulates DMP1 expression during odontoblastic differentiation." Journal of dental research98.5 (2019): 597-604).  

Hu, H.-M.; Mao, M.-H.; Hu, Y.-H.; Zhou, X.-C.; Li, S.; Chen, C.-F.; Li, C.-N.; Yuan, Q.-L.; Li, W. Artemisinin Protects DPSC from Hypoxia and TNF-α Mediated Osteogenesis Impairments through CA9 and Wnt Signaling Pathway. Life Sciences 2021, 277, 119471.

Chmilewsky, F.; About, I.; Chung, S.H. C5L2 Receptor Represses Brain-Derived Neurotrophic Factor Secretion in Lipoteichoic Acid-Stimulated Pulp Fibroblasts. J Dent Res 2017, 96, 92–99.

  1. To synchronize all the cells to the same cell cycle phase, cells need to be starved before treatment. In the current study, cells were treated in a complete regular media and did not starve??

Response: DPSCs were examined at early, mid, and late phases of odontoblastic differentiation. Instead of starving the cell, we wanted to see the comparison of the process within normal possible environment. Also, the point of using dentinogenic media is to give nutrients during the odontoblastic differentiation. And thanks for the suggestion, we definitely keep in mind next time.

  1. In the results sections, please don’t repeat the methods.

Response: Thank you for your suggestion. Results have been modified accordingly.

  1. In Fig.2, S. Why authors used resting rather than control.

Response: Thank you for noticing the error. It has been corrected with control.

Once again, we are thankful and appreciate your efforts to evaluate our study; it really helped us improving the quality of our manuscript. We do hope that concerned queries have been answered accordingly, and the revised version of manuscript is acceptable.

Thank you for anticipation.

Round 2

Reviewer 1 Report

Dear authors,

I really appreciate the corrections made based on my guidance on the article "BDNF/TrkB is a crucial regulator in the differentiation of inflammation-mediated odontoblastic DPSCs"

In this form, with the appropriate modifications I would accept this work.

Thank you

Reviewer 2 Report

The authors addressed most of the reviewer's comments.

The English language of the manuscript is written well.